# Intake of Nutrient and Non-Nutrient Dietary Antioxidants. Contribution of Macromolecular Antioxidant Polyphenols in an Elderly Mediterranean Population

**DOI:** 10.3390/nu11092165

**Published:** 2019-09-10

**Authors:** Isabel Goñi, Ana Hernández-Galiot

**Affiliations:** Department Nutrition and Food Science, Faculty of Pharmacy, University Complutense of Madrid, Spain Research Group, Nutrition and Gastrointestinal Health, 28040 Madrid, Spain

**Keywords:** bioactive components, dietary antioxidants, macromolecular polyphenols, dietary fiber, elderly

## Abstract

The intake of antioxidants in the diet is a useful parameter to estimate the potential of diet to prevent chronic diseases associated with oxidative stress and ageing. The objective was to estimate the intake of nutrient and non-nutrient antioxidants associated with the dietary fiber matrix in a healthy and functionally independent population aged over 80, estimating the intake of antioxidant nutrients and including soluble low molecular weight and macromolecular polyphenols in the non-nutrient antioxidant group. Specific nutrients related to oxidative stress (copper, zinc, selenium, manganese, vitamins A, C and E) were ingested in optimal quantities according to reference values. Total intake of non-nutrient antioxidants was 2196 mg/person/day, and macromolecular polyphenols were found to be the main dietary antioxidants, contributing 71% to the total intake of phenolic compounds. The intake, metabolism and physiological effects of all nutrient and non-nutrient dietary antioxidants must therefore be taken into account when evaluating their health benefits.

## 1. Introduction

Numerous studies have shown that the elderly have increased oxidative stress and impaired antioxidant defense systems [1,2], which appears to be a contributory factor for neurological damage, dementia and depression and to be responsible for the onset and progression of chronic degenerative diseases [3] such as diabetes [4], atherosclerosis [5] and hypertension [6]. Both preventive and chain breaking antioxidants have a role in limiting the oxidative stress that accompanies ageing and disease.

The intake of dietary antioxidants could be a useful parameter for estimating the potential of diet to prevent chronic diseases associated with oxidative stress and ageing. However, there is limited research and data on the antioxidant status of healthy elderly people.

In a balanced healthy diet, plant foods including fruits, vegetables, legumes, grains, nuts, red wine, tea, olive oil, herbs and spices and beverages provide a significant amount and variety of antioxidants associated with a lower risk of chronic diseases [7,8]. Dietary antioxidants are complex mixtures of hundreds of compounds. Some are micronutrients such as vitamins A, C and E and minerals such as copper, zinc or selenium. Other antioxidants are bioactive phytochemicals such as phenolic compounds. In fact, phenolic compounds are frequently used as anti-ageing compounds in both dietary supplements and cosmetic formulations [9,10]. Phenols are well known for their beneficial effects on human health due to their antioxidant, cardioprotective, anticancer, anti-inflammatory and antimicrobial properties. In addition, it has been demonstrated that phenols may also prevent neurodegenerative diseases and aging [11,12]. The estimation of the intake of vitamins and minerals in the diet is relatively frequent, but antioxidant phytonutrients are not often evaluated in the diet of the elderly. Polyphenols are quantitatively the main dietary antioxidants in a total diet [13]. Several descriptive papers on intakes of flavonoids and other polyphenols have been published using either the USDA databases [14], the Phenol-Explorer database [8] or custom databases [13].

As is known, foods contain a wide variety of phenolic compounds that can be classified in physiological terms according to their solubility in the intestinal medium. Most of them are associated with the components of the fiber matrix to both soluble and insoluble fraction. Fiber and associated compounds synergistically interact with the colonic microbiota by generating active metabolites, many of them with antioxidant activity [15]. Some may be solubilized in the stomach and small intestine and can be at least partially absorbed through the small intestinal mucosa, producing metabolic and systemic effects. These polyphenols are low molecular weight compounds (low molecular polyphenols, LP), partially available in the small intestine [13,15]. Only between 5% and 10% of the bioaccessible polyphenols in the small intestine are bioavailable [16], so a large part of the polyphenols that are potentially bioaccessible in the small intestine may reach the colon due to their low bioavailability [13].

Other phenolic compounds appear as high molecular weight structures (macromolecular polyphenols, MP) formed by polymeric polyphenols or by low molecular weight polyphenols associated with other compounds such as polysaccharides or proteins [17]. The main macromolecular polyphenols are proanthocyanidins (MPP) and hydrolysable polyphenols (MHP). LP and MP exhibit different physiological properties, but both exert antioxidant activity [17], so when they reach the colon they interact with microbiota and yield bioavailable and active metabolites. They also remain in the colon longer than other colonic substrates, thereby increasing the time that beneficial metabolites will be in contact with colonic cells and microbiota, and their circulation time through the human body once absorbed [18,19]. Polyphenolic compounds are therefore considered components of dietary fiber [18].

The objective of this study was to advance the knowledge of antioxidant intake in the diet, in a healthy and functionally independent elderly population aged over 80, estimating the intake of nutrient and non-nutrient antioxidants, including in this last group the soluble and macromolecular antioxidants consumed in the total diet.

## 2. Materials and Methods

### 2.1. Procedure and Patient Characteristics

A cross sectional survey—Garrucha Older Health Study—was conducted in very old women and men living in Garrucha (8626 registered inhabitants) in Almería (Spain), located on the Mediterranean coast. All non-institutionalized inhabitants (n = 464) aged 80 and over and registered in the municipal census in 2016 were invited by letter to participate in the study. The final sample comprised 109 participants (Women = 60, Men = 49) aged between 75 and 93 years. They were mentally competent and functionally independent to perform both basic and instrumental daily activities, and had no cognitive problems. Functional assessment was evaluated using the Barthel [20] and Lawton and Brody indices [21], the results of which were not included in this study [22].

Data were collected by interview, using comprehensive geriatric and nutritional assessment. Interviews were carried out in the respondents’ homes by researchers trained in geriatric assessment and nutritionists. All the subjects gave their informed consent for inclusion before taking part in the study.

The study was conducted in accordance with the Declaration of Helsinki, under a collaboration agreement between the Complutense University in Madrid and the Garrucha City Council, and was approved by the Ethics Review Board of the Complutense University in Madrid (16/165E). 

### 2.2. Food Consumption

Food consumption data were collected in person by trained dietitians using a standardized interview based on an overall questionnaire incorporating socio-demographic status and lifestyle factors, three 24-h recalls and a validated semi-quantitative food frequency questionnaire [23]. Food consumption was expressed as g (or mL)/person/day. Recipes were separated according to their ingredients. Daily intake of vitamins (A, C and E) and minerals (copper, selenium, zinc, and manganese) were estimated using a computer program [24]. Values were expressed as mean ± standard deviation.

### 2.3. Sample Preparation

The edible portion of the daily amount consumed per capita for each plant food as eaten was weighed and grouped into five samples, one for each of the five types of plant foods: cereals (total: 138.6 g), vegetables (total: 354.2 g), legumes (total: 19.92 g), nuts (total: 2.3 g) and fruits (total: 347.2). These five samples corresponded to the total per capita daily intake of solid plant foods in the study population. Each duplicated sample was freeze-dried, ground and stored until analysis. Beverages (total: 300.3 mL) and vegetable oils (total: 22.3 mL) were analyzed individually. Distilled drinks were not included in the beverage group (Table 1).

### 2.4. Adherence to the Mediterranean Diet

Adherence to the Mediterranean diet was determined by the MEDAS screener developed in the PREDIMED study [25]. A personal interview was conducted with each participant to complete a questionnaire consisting of 14 questions. The 14-item MEDAS screener includes 12 items with targets for food consumption, and another two items with targets for food intake habits characteristic of the Mediterranean diet, in order to determine whether the respondent consumes olive oil and the amount ingested daily.

Each question scored 0 or 1. One point was given for each target achieved. If the condition was not met, 0 points were recorded for the category.

The questions relate to the type of fat used for cooking, number of daily or weekly fruit servings, vegetables, legumes, nuts, fish, soft drinks and wine, as well as the frequency of consumption of sweets and salads, and the preference for consuming certain types of meat and meat products. The total MEDAS score ranges from 0 to 14, with a higher score indicating a better adherence to the Mediterranean diet. A MEDAS score of ≥7 (mid-range value) represented a modest adherence, and a score of ≥9 represented strict adherence to a healthy dietary pattern [26].

### 2.5. Determination of Phenolic Compounds

Plant food samples and vegetable oils were previously extracted following the methodology described by Saura-Calixto and Goñi (2006) [27]. Briefly, solid samples were sequentially extracted at room temperature with a solution of methanol/water in acid medium and acetone/water. Each solid sample (0.5 g) was placed in a capped centrifuge tube, and 20 mL of methanol/water solution (50:50, *v*/*v*) acidified at pH 2 with HCl was added. The tube was thoroughly shaken at room temperature for 1 h and was centrifuged at 2500 × *g* in a Thermo Heraeus Megafuge 11 (Thermo Fisher, Waltham, MA, USA) for 10 min and the supernatant was recovered. A total of 20 mL acetone/water (70:30, *v*/*v*) was added to the residue, and shaking and centrifugation were repeated. The methanol and acetone extracts were combined and used to determine LP polyphenols by the Folin–Ciocalteau procedure [28]. This method has been used regularly in similar samples [13]. Gallic acid was used as standard (calibration curve: y = 3.628x + 0.0651; *R*^2^ = 0.9958) and results were expressed as mg of gallic acid equivalents. Phenolic compounds were measured directly on the original beverages.

Residues from the double extraction were subjected to two different previously reported procedures [13] in order to obtain the two main constituents: MHP and MPP or condensed tannins. Vegetable oils were extracted with a previously reported specific extraction using methanol [29].

### 2.6. Statistical Analysis

Extraction and analytical procedures were performed in triplicate. Results were expressed as mean values ± standard deviation, on a dry matter basis. Linear regression analysis between variables was performed with the SPSS 21 statistical package for Windows software. Correlation between variables was considered to be statistically significant when *p* < 0.05.

## 3. Results

The average age of the study population was 81 years and all individuals showed a high adherence to the Mediterranean diet pattern according to the MEDAS values. Other interesting results in relation to lifestyle were that none smoked—although some had smoked during certain periods in their life—and that most were sedentary. The energy intake was in accordance with the recommended levels for the Spanish population, and were even lower in some cases. Mean BMI value did not differ between women and men (Table 2).

### 3.1. Intake of antioxidant nutrients

A deficit of vitamin intake is common in the elderly population, as is the use of dietary supplements. This was not the case in this study because the state of health was good and the intake of specific nutrients related to oxidative stress also proved to be adequate (Table 3). Although some participants were medicated due to the presence of pathologies, none consumed dietary supplements. Only a slight deficit was observed in vitamin A intake in the case of men, and vitamin E in both men and women. However, vitamin E intake was higher than the average for Spain in the over 64 group, but was slightly lower than the recommended figure (Table 3). Our results also indicate that vitamin C intake greatly exceeded the recommended figures, as is common in Mediterranean countries.

The minerals involved in antioxidant mechanisms were ingested in sufficient quantities, except in the case of zinc (Table 3).

### 3.2. Intake of non-nutrients antioxidants 

Total phenolic compounds including soluble and macromolecular polyphenols were measured in plant foods and beverages consumed by elderly individuals. The mean total polyphenol intake was 2196 mg, of which only 29% were LP (Figure 1). The MP group combines MHP and MPP. Cereals, fruits, vegetables and legumes were the main contributors to MHP, (Figure 2), while the main contributors to MPP intake were fruits and legumes. Fruits were major contributors to the total intake of polyphenols, but only 22.5% corresponded to LP. It was notable that the main contributor to MPP intake was certain food groups which had the lowest daily consumption such as legumes and nuts, but with a high MPP content. This fact was more relevant for legumes, as they contained 14.7% of phenolic compounds on a dry basis, of which 89% were MP (Table 4).

## 4. Discussion

This study has two notable characteristics: (1) the participants are octogenarians who live in their own home and are functionally independent. They have lived their entire life or most of it in a Mediterranean region, which has determined their lifestyle. Studies in this age range are scarce and often conducted on institutionalized individuals. It should also be noted that age-related changes, such as sensory perception, decreased sensitivity to thirst, digestive and cognitive function, quality of life, etc., can interfere with nutritional status, and these characteristics obviously worsen with advancing age. An important part of the study was, therefore, to determine eating habits, physical activity and lifestyle in order to assess nutritional status; and (2) the oxidative stress theory of ageing explains ageing at the molecular level and has been used for several years together with longevity or lifespan to study ageing [32]. Increased reactive oxygen and nitrogen species levels can affect the cell processes that limit lifespan, as a gradual increase in oxidative damage intensifies the stimulation of the stress response and gradually increases the generation of free radicals and age-dependent diseases. The amounts of antioxidants available in the diet are therefore related to the health status of the elderly population. We quantified the intake of the main antioxidant nutrients (vitamins and minerals) and non-nutrient antioxidants (phenolic compounds), including soluble polyphenols in the intestinal environment (LP) and polymeric polyphenols (MP). MP and most of LP are compounds associated with the fiber matrix and the important physiological effects derived from them are often attributed to the components of the fiber matrix [18]. Although studies quantifying LP are common, MP are often ignored. However, both LP and MP have a potent antioxidant activity [17] and are (or should be) ingested in the regular diet.

It is notable that the current MEDAS score for the Spanish adult population is about 6.3 [26], which is significantly lower than the value found in this study (Table 2). This positive difference seems to be determined by the participants’ lifestyle, as they followed a healthy diet that corresponded to the Mediterranean dietary pattern [26]. They may have learned healthy habits during childhood which they maintained throughout their lives, pointing to the importance of nutritional education, which should be emphasized. Energy intake remained at recommended values which may positively affect their health status. A lower caloric intake affects body weight control and also tends to attenuate the process of cell damage with ageing, with reduced lipid peroxidation, lower accumulation of oxidized proteins and oxidative DNA damage [33]. As can be seen in Table 2, the BMI (Body Mass Index) values could be considered elevated for the adult population, but are not a problem for the health of older people, since no association has been found between excess weight and increased risk of mortality [34]. According to other authors, the highest quality-of-life scores after age 70 were categorized for more restrictive BMI values between 24 and 30 kg/m^2^ [35]. All the results obtained in this study were within this range, so we cannot say that the participants were overweight. 

### 4.1. Intake of Antioxidant Nutrients

The ingestion of some nutrients may be very effective in increasing antioxidant defenses by up-regulating the activity of the antioxidant enzymes that are normally present in cells. There are several enzyme systems within the body that effectively scavenge free radicals. Cells also have many low molecular weight antioxidants. Several vitamins and micronutrients are active in quenching free-radical species or are required as cofactors for antioxidant enzymes [36]. Their intake has been assessed in this work using European [30] and Spanish [31] dietary guidelines as reference values, which have the same values for all ages from 60. This was a limitation of the study, as dietary guidelines should be adjusted every five or ten years after 60, although as this is not commonly done, updated values are not available.

The intake of antioxidant nutrients was right in this study. It is important to note that the study population was located in the Mediterranean region, where the consumption of fresh fruits and vegetables is a feature of the dietary pattern. A wide variety of freshly caught fish and fish products are available in this area, along with several crops per year of various fruits and vegetables. Fruits and vegetables are key sources of a number of essential nutrients (vitamins and minerals) and other bioactive substances known as phytochemicals or phytonutrients [5].

Among the significant antioxidant nutrients are vitamins C and E. Considerable biochemical and physiological evidence suggests that ascorbic acid functions as a free radical scavenger and inhibits the formation of potentially carcinogenic N-nitrous compounds from nitrates [37]; the consumption of citrus fruit has been shown to reduce the risk of death in free-living elderly individuals [38]. 

Vitamin E is a generic term for all tocopherols and their derivatives with biological activity. Numerous epidemiological studies have shown that tocopherols induce a protective effect against oxidative stress [39], preventing or delaying the development of degenerative and inflammatory diseases [40]; a high vitamin E intake is related to a low risk of age-related and chronic diseases [37]. 

The intake of minerals involved in antioxidant mechanism coincided with the average intake of the Spanish population, although they were lower than the reference values. Despite this, no diseases related to Zn deficiency were detected. A possible explanation is that current zinc recommendations are high and should be revised. It is worth noting the high selenium intake in both men and women. Selenium is a key component of several functional seleno-proteins that protect tissues and membranes from oxidative stress and control the cell redox status [41].

### 4.2. Intake of Non-Nutrient Antioxidants

Phenolic compounds are quantitatively the main dietary non-nutrient antioxidants [17,27] and are related to the antioxidant hypothesis regarding the beneficial effects of plant foods that can be attributed to the synergy or interactions of antioxidant bioactive compounds and other nutrients in the diet as a whole. Therefore, polyphenols are important constituents of dietary fiber and confer specific properties, depending on the amount, the type of compounds and bioavailability [15]. 

Extensive knowledge of the polyphenol content of foods and diets is essential for studies of nutrition and health, but only a fraction of these are considered in current research, because the literature and databases on food composition focus almost exclusively on LP, and often ignore MP. However, from a qualitative point of view, MP intake is very important for health, as MP may be fermented by microbiota and release bioactive metabolites which, in combination with their parent compounds, either have an in situ local effect or are absorbed and exhibit potential systemic effects [15].

MP intake is also of significant interest from a quantitative point of view, as the limited published data indicate that MP are widely present in the human diet, and that MP intake may even be higher than LP intake [13]. Despite their unquestionable qualitative and quantitative importance, though, they are often overlooked in nutritional studies [8,14].

The intake of phenolic compounds was similar to the results obtained in a previous work on the Spanish population a few years ago [13]. The main contributors to the total polyphenol intake were MP (71% vs. 29% from LP). This is a new aspect that arises from these data, as MP were not previously considered in the determination of polyphenol intake, although they are responsible for over half of daily dietary polyphenols. 

A recent work deriving from the EPIC study (European Prospective Investigation into Cancer and Nutrition) [14] provides an important and detailed description of polyphenol intake in a large European multicenter study. However, the work focuses only on LP, and references a total intake of phenolic compounds ranging from 500 to 2000 mg/person/day. These amounts are obviously lower than the estimated total intake of polyphenols in this work, but may correspond to the intake of LP (634 mg/p/d, Figure 1). These data also agree with results found by other authors [8] who estimated a mean total polyphenol intake of 820 mg/d calculated based on the Phenol-Explorer database [42].

The scarcity of available research, including data on MP, was a limitation when comparing the results with other populations and drawing conclusions relating to health and the intake of polyphenolic compounds. Another related limitation regarding food consumption data should also be taken into account. This work was based on the current consumption of foods registered in at least three 24h-recall questionnaires from each participant, while other studies refer to phenolic intake based on the frequency of consumption of food groups from household spending questionnaires. The results could therefore be indicative, but not entirely comparable.

## 5. Conclusions

In summary, this study includes data on the consumption of antioxidant nutrients and emphasizes the importance of non-nutrient antioxidants associated with dietary fiber, because they confer properties with potential effects on promoting gastrointestinal health. The daily diet of the elderly group in the study supplies large amounts of polyphenolic compounds, LP and MP (2196 mg), with a majority of MP (71%) most of them, associated with the fiber matrix. The intake, metabolism and physiological effects of all nutrient and non-nutrient dietary antioxidants must therefore be taken into account when evaluating their health benefits. 

## Figures and Tables

**Figure 1 nutrients-11-02165-f001:**
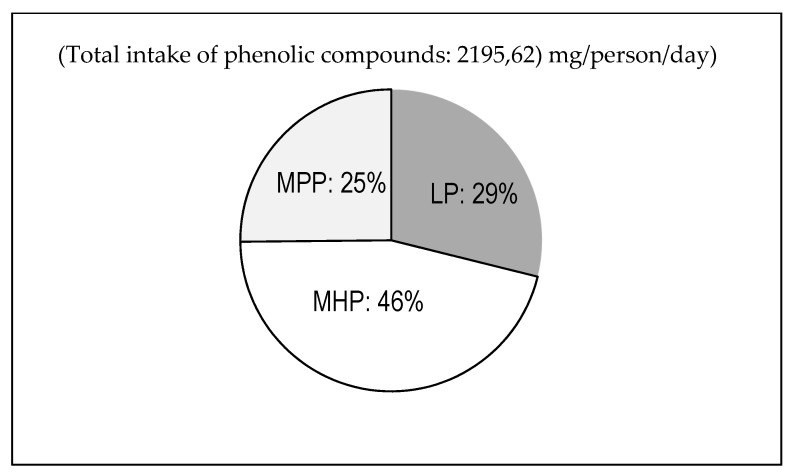
Contribution of macromolecular polyphenols to the daily intake (LP: Low molecular polyphenols; MHP: Molecular hydrolysable polyphenols; MPP: Macromolecular polymeric polyphenols).

**Figure 2 nutrients-11-02165-f002:**
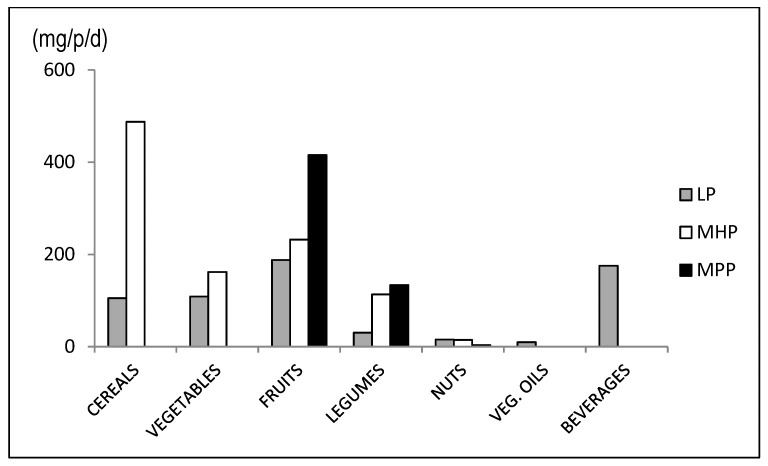
Contribution of plant foods to the intake of phenolic compounds in the total diet (LP: Low molecular polyphenols; MHP: Molecular hydrolysable polyphenols; MPP: Macromolecular polymeric polyphenols).

**Table 1 nutrients-11-02165-t001:** Consumption of plant foods, oils and beverages in an elderly Mediterranean population aged over 80.

Plant Foods		g Fresh Matter/Day	g Edible Portion/Day
Cereals (g)	White bread (47.65%), rice (13.61%), muffin (11.93%), pasta (7.95%), biscuit (6.89%), whole-grain bread (3.82%), corn (2.13%), cornflakes (1.94%), flour (0.86%), others (3.22%)	138.64	138.64
Vegetables (g)	Tomato (18.90%), green bean (13.92%), lettuce (10.35%), onion (5.61%), carrot (4.45%), zucchini (3.89%), cucumber (3.78%), pepper (3.44%), asparagus (2.04%), leek (1.63%), mushroom (1.61%), aubergine (1.23%), others (29.15%)	354.20	300.81
Fruits (g)	Apple (17.78%), orange (14.51%), pear (11.10%), cantaloupe (10.69%), banana (8.49%), tangerine (6.21%), plum (5.68%), fig (5.19%), watermelon (5.13%), peach (4.84%), others (10.37%)	347.16	267.47
Legumes (g)	Lentil (52.14%), chickpea (37.45%), white bean (10.40%)	19.92	17.85
Nuts (g)	Walnuts (48.55%), almond (39.41%), pistachio (12.04%)	2.31	2.31
Vegetable oils (mL)	Olive oil extra virgin (69.82%), olive oil (28.90%), olive oil virgin (0.76%), sunflower oil (0.52%)	22.26	22.26
Beverages (mL)	Natural juice (18.13%), red wine (13.33%), decaffeinated coffee (13.27%), refreshments (12.70%), beer (11.87%), commercial juice (9.22%), coffee (7.85%), infusion (6.88%), others (6.75%)	300.33	300.33

**Table 2 nutrients-11-02165-t002:** General characteristics of the participants ^1^.

	Total	Women	Men	*p-*Value ^2^
Age (years)	80.98 ± 4.58	81.96 ± 4.47	80.16 ± 4.58	0.316
Mediterranean Diet Adherence Screener	9.33 ± 1.51	9.31 ± 1.35	9.35 ± 1.66	0.908
Energy intake (Kcal/day)	1569 ± 337	1424 ± 314	1682 ± 314	0.864
Body Mass Index (Kg/m^2^)	27.85 ± 6.48	27.49 ± 8.15	28.17 ± 4.71	0.697
Number of Diseases	6.19 ± 4.07	7.38 ± 4.22	5.19 ± 3.72	0.042
Number of drugs	5.05 ± 3.01	4.42 ± 3.07	5.58 ± 2.91	0.150
Tobacco consumption	No	No	No	
Physical Activity (min/person/day)	55.10 ± 55.22	29.45 ± 26.55	76.61 ± 63.68	0.001

^1^ Data are presented as mean ± standard deviation. ^2^
*p* ≤ 0.05, significant difference between men and women.

**Table 3 nutrients-11-02165-t003:** Intake of antioxidant nutrients in the free-living elderly population over 80.

	Participants (>80 years)		Intake in Spain [30] (>64 years)		Reference Values [31] (>60 years)	
	Men (mean ± SD)	Women (mean ± SD)	men	women	Men	Women
Copper (mg)	1.32 ± 1.24	1.09 ± 0.39	1.1–1.9	0.9–1.9	1.0–1.5	
Zinc (mg)	7.72 ± 2.24	7.47 ± 1.95	9.4 ± 1.4	7.8 ± 1.2	15	15
Selenium (µg)	93.43 ± 20.09	75.97 ± 20.60	-	-	70	55
Manganese (mg)	3.18 ± 1.23	2.79 ± 0.65	-	-	2.2–4.9	2.4–4.4
Vitamin A ^1^ (mg)	0.80 ± 0.35	0.77 ± 0.29	0.5 ± 0.1	0.4 ± 0.0	1.0	0.8
Vitamin C (mg)	137.88 ± 56.09	131.67 ± 63.65	126 ± 50	115 ± 38	60	60
Vitamin E ^2^ (mg)	9.77 ± 3.65	8.48 ± 2.77	8.4 ± 2.3	7.5 ± 1.7	12	11

^1^ Expressed as retinol equivalents. ^2^ Expressed as α-tocopherol equivalents.

**Table 4 nutrients-11-02165-t004:** Phenolic compound content of plant foods in the diet consumed by an elderly population aged over 80 (mg/g original dry sample, mean ± standard deviation).

Food Group	LP	MHP	MPP	TOTAL
Cereals	1.01 ± 0.03	4.67 ± 0.59	nd	5.68
Vegetables	2.95 ± 0.13	4.39 ± 0.42	nd	7.34
Fruits	5.60 ± 0.19	6.91 ± 0.55	12.36 ± 1.04	24.87
Legumes	1.62 ± 0.12	6.02 ± 0.31	7.09 ± 0.53	14.73
Nuts	8.72 ± 0.33	8.37 ± 0.58	1.88 ± 0.17	18.97
Vegetables oils	44.38 ± 1.27 (mg/100 mL)	-	-	44.38
Beverages	58.49 ± 1.84 (mg/100 mL)	-	-	58.49

LP: Low molecular polyphenols; MHP: Molecular hydrolysable polyphenols; MPP: Macromolecular polymeric polyphenols; nd: not detected

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
