# Peer review of "Intake of Nutrient and Non-Nutrient Dietary Antioxidants. Contribution of Macromolecular Antioxidant Polyphenols in an Elderly Mediterranean Population"

_nutrients, 2019, doi:10.3390/nu11092165_

Round 1

Reviewer 1 Report

- The authors should improve Introduction Section and use recent references.

For instance, regarding phenolic compounds, the authors reported that “Other antioxidants are bioactive phytochemicals such as phenolic compounds. In fact, phenolic compounds are frequently used as anti-ageing compounds in both dietary supplements and cosmetic formulations.” (lines 39-41)

The authors should better discuss the wide range of beneficial effects of the phenolic compounds. For example, they could use the following paragraph: “phenols are well known for their beneficial effects on human health due to their antioxidant, cardioprotective, anticancer, anti-inflammatory and antimicrobial properties. In addition it has been demonstrated that phenols may also prevent neurodegenerative diseases and aging (Di Mauro et al., 2017 ; Li et al., 2014; Fraga et al., 2019; Khurana et al., 2013; Rossi et al., 2008)”. In addition, the authors could cite some works about the use of phenols in cosmetic field (Rodrigues et al., 2015).

- Di Mauro et al., Polyphenolic profile and antioxidant activity of olive mill wastewater from two Sicilian olive cultivars: Cerasuola and Nocellara etnea, Eur. Food Res. Technol., 2017, 243, 1895–1903.

- Li et al., Resources and biological activities of natural polyphenols. Nutrients, 2014, 6:6020–6047

- Fraga et al.,The effects of polyphenols and other bioactives on human health. Food Funct. 2019 10, :514-528.

- Khurana et al, Polyphenols: benefits to the cardiovascular system in health and in ageing. Nutrients, 2013, 5:3779–3827

- Rossi et al., Benefits from dietary polyphenols for brain ageing and Alzheimer’s disease. Neurochem Res, 2008, 33:2390–2400

- Rodrigues et al., Coffee silverskin: a possible valuable cosmetic ingredient., Pharm Biol., 2015, 53, 386-94.

- Regarding Materials and Methods Section, the authors should report the equation of gallic acid calibration curve (lines 132-133). In addition they affirm that “The study was conducted in accordance with the Declaration of Helsinki under a collaboration agreement between the Complutense University in Madrid and the Garrucha City Council, and was approved by the Ethics Review Board of the Complutense University in Madrid”. Generally the number of document that approves the research is reported, so the authors could include this information.

- The authors should improve the discussions of results.

The authors reported that “A lower caloric intake also tends to attenuate the process of cell damage with ageing, with reduced lipid peroxidation, lower accumulation of oxidized proteins and oxidative DNA damage, and also improves control of body weight since the participants were sedentary”. (lines 173-176).

The authors should discuss the importance of physical exercise in the elderly (Tomasello et al., 2019; Biochimica clinica, doi 10.19186/BC_2019.044; Tomasello et al., 2017, Oncol Lett., 13, 441–448.)

Author Response

Response to Reviewer 1 Comments

Point 1: The authors should improve Introduction Section and use recent references.

For instance, regarding phenolic compounds, the authors reported that “Other antioxidants are bioactive phytochemicals such as phenolic compounds. In fact, phenolic compounds are frequently used as anti-ageing compounds in both dietary supplements and cosmetic formulations.” (lines 39-41)

Response: Two references related to the use of phenolic compounds in cosmetics have been introduced

Point 2: The authors should better discuss the wide range of beneficial effects of the phenolic compounds. For example, they could use the following paragraph: “phenols are well known for their beneficial effects on human health due to their antioxidant, cardioprotective, anticancer, anti-inflammatory and antimicrobial properties. In addition it has been demonstrated that phenols may also prevent neurodegenerative diseases and aging (Di Mauro et al., 2017 ; Li et al., 2014; Fraga et al., 2019; Khurana et al., 2013; Rossi et al., 2008)”. In addition, the authors could cite some works about the use of phenols in cosmetic field (Rodrigues et al., 2015).

Response: The paragraph suggested by the reviewer and the reference to a recent review on the beneficial effects of phenolic compounds have been included

Point 3: Regarding Materials and Methods Section, the authors should report the equation of gallic acid calibration curve (lines 132-133). In addition they affirm that “The study was conducted in accordance with the Declaration of Helsinki under a collaboration agreement between the Complutense University in Madrid and the Garrucha City Council, and was approved by the Ethics Review Board of the Complutense University in Madrid”. Generally the number of document that approves the research is reported, so the authors could include this information.

Response: Information on the use of gallic acid in the calibration curve for the determination of polyphenolic compounds and the reference of the approval by the Ethic Committee have been included

Point 4: The authors should improve the discussions of results.The authors reported that “A lower caloric intake also tends to attenuate the process of cell damage with ageing, with reduced lipid peroxidation, lower accumulation of oxidized proteins and oxidative DNA damage, and also improves control of body weight since the participants were sedentary”. (lines 173-176).

The authors should discuss the importance of physical exercise in the elderly (Tomasello et al., 2019; Biochimica clinica, doi 10.19186/BC_2019.044; Tomasello et al., 2017, Oncol Lett., 13, 441–448.)

Response: This paragraph has been slightly changed in order to clarify the meaning.

The articles indicated by the reviewer are interesting, but they refer to breast cancer survivors, while the participants in this study are healthy people.

We believe that delving into these aspects would turn attention away from the objectives of the work and increase the size of the manuscript too much.

Reviewer 2 Report

Dear Authors,

            The manuscript "Intake of nutrient and non-nutrient dietary antioxidants. Contribution of macromolecular antioxidant polyphenols in an elderly Mediterranean population.” by the authors Isabel Goñi* and Ana Hernández-Galiot is dealing with the analysis of antioxidant intake by  population aged over 80. In my opinion it is very important topic as influence of diet on population health is nowadays a great problem, especially in group of elderly, more susceptible on oxidative stress related to age

I read the article with great interest, the manuscript is very interesting  and  expands knowledge about the different polyphenol groups intake in elderly people

My remarks are summarized as follows:

Line 42 – “is” is written  by larger font than rest of text,

Lines 98-101- the description of samples is not quite clear for me. Are they a result of interviews and according to interviews , samples were calculated? (quantitatively and qualitatively)

Lines 124-126 -  „solid samples were sequentially  extracted at room temperature with a solution of methanol/water in acid medium  and acetone/water.” is written  by larger font than rest of text

Line 127 – “acidic methanol/water/HCl (50:50, v/v; pH 2)” – should be better described – was HCL added to achieve pH 2?

Lines 250-252 and 272-274 – a  figure captions should be directly under  figure, no on other page

Figure 2 -  for MPP there are black color on graph but no in legend

Author Response

Response to Reviewer 2 Comments

Point 1: Line 42 – “is” is written  by larger font than rest of text,

Response: has been corrected.

Point 2: Lines 98-101- the description of samples is not quite clear for me. Are they a result of interviews and according to interviews , samples were calculated? (quantitatively and qualitatively)

Response: Food consumption data was collected through questionnaires (24-hours recalls and a semi-quantitative food frecuency questionnaire). Based on these data, the corresponding foods were purchased and the edible portions of all of them were grouped into the 7 samples (cereals, vegetables, fruits, nuts, legumes, vegetable oils and non-alcoholic beverages). Analytical determinations were made in these samples.

Point 3: Lines 124-126 -  „solid samples were sequentially  extracted at room temperature with a solution of methanol/water in acid medium  and acetone/water.” is written  by larger font than rest of text

Response: has been corrected.

Point 4: Line 127 – “acidic methanol/water/HCl (50:50, v/v; pH 2)” – should be better described – was HCL added to achieve pH 2?

Response: has been clarified.

Point 5: Lines 250-252 and 272-274 – a  figure captions should be directly under  figure, no on other page

Response: has been corrected.

Point 6: Figure 2 -  for MPP there are black color on graph but no in legend

Response: has been corrected.

Round 2

Reviewer 1 Report

I don’t believe the manuscript has been significantly improved. The authors affirm in their response to reviewer 1 that have added some informations but it is not true (see my comments at point 3 and 4).

- Point 3: The authors have reported that ”Information on the use of gallic acid in the calibration curve for the determination of polyphenolic compounds and the reference of the approval by the Ethic Committee have been included”.

Response: the authors have not reported the equation of gallic acid calibration curve in the revised manuscript. In my opinion, they must report this information, including R2 value.

-Point 4: The authors have reported that the articles indicated by the reviewer refer to breast cancer survivors, while the participants in this study are healthy people.

Response: This is not true. The first suggested article represents a study performed on healthy elderly subjects (Tomasello et al., 2019; Biochimica clinica, doi 10.19186/BC_2019.044). So they could have cited this paper.

Finally, regarding the Point 2: The paragraph suggested by the reviewer on the beneficial effects of phenolic compounds have been included, but they have not included the references suggested.

In particular, the paragraph is similar to the one cited in the work (Di Mauro et al., 2017). In my opinion the authors have to include the suggested works or they have to modify the paragraph. If they don’t it could be considered plagiarism.

In addition, I would like to evidence that I have suggested to modify references section because of there are about 26 references published before the year 2015.

Author Response

The authors appreciate the comments and suggestions of Reviewer 1.

Point 2: The paragraph suggested by the reviewer on the beneficial effects of phenolic compounds have been included, but they have not included the references suggested.

Response:

The paragraph introduced in response to the reviewer's suggestion refers to a very abbreviated description of the general effects of phenolic compounds, which can be found described with more or less similarity in many texts. Therefore, when the reviewer suggested introducing this text, we decided to include the reference suggested by the revisor (Di Mauro et al., 2017) and another reference of a good current revision (Fraga et al, 2019).

We made a mistake and put an incorrect reference from the same author. This error has been corrected.

Point 3: The authors have not reported the equation of gallic acid calibration curve in the revised manuscript. In my opinion, they must report this information, including R2 value

Response:

The equation was not included because the Academic Editors indicated another posible option:

"- Include information regarding the gallic acid calibration curve (concentration range and R2 value) used for the determination of phenolic compounds. Alternatively, the authors may refer to a publication from their laboratory dealing with this analytical procedure.".

The authors chose the other option proposed by the Academic Editors.

However, in response to the reviewer's comments, the equation and the value of R2 have been included.

Point 4: This is not true. The first suggested article represents a study performed on healthy elderly subjects (Tomasello et al., 2019; Biochimica clinica, doi 10.19186/BC_2019.044). So they could have cited this paper.

Response:

We apologised to the reviewer for the confusion, as the authors were referring to article of Tomasello et al., 2017, while the reviewer referred to the article of Tomasello et al., 2019.

The importance of physical activity in the elderly is a very interesting topic, but it is not an objective of this work, so to include the reference of Tomasello et al, 2019, we would have to address the discussion of more results not included in this work and we believe that delving into these aspects would turn attention away from the objectives of the work and increase the size of the manuscript too much, as we indicated in our previous response.
